# The Role of Prediction Error in 4-Year-Olds' Learning of English Direct Object Datives

Chiara Gambi [1,2,*] and Katherine Messenger [1]

1   Department of Psychology, University of Warwick, Coventry CV4 7AL, UK; k.messenger@lancaster.ac.uk
2   School of Psychology, Cardiff University, Cardiff CF10 3AT, UK
*   Correspondence: chiara.gambi@warwick.ac.uk

**Abstract:** Is children's acquisition of structural knowledge driven by prediction errors? Error-driven models of language acquisition propose that children generate expectations about upcoming words (prediction), compare them to the input, and, when they detect a mismatch (i.e., prediction error signal), update their long-term linguistic knowledge. But we only have limited empirical evidence for this learning mechanism. Using a novel touch-screen app and a pre-post training between-subjects design, we tested the effect of prediction errors on 120 English-learning 4-year-olds' understanding of challenging direct object datives. We hypothesized that children who are exposed to input that encourages the generation of prediction error signals should show greater improvements in their post-test comprehension scores. Consistent with error-driven models of language learning, we found that children exposed to sentences that encouraged the generation of incorrect linguistic predictions improved numerically more than those who were exposed to sentences that did not support predictions. However, we caution that these preliminary findings need to be confirmed by additional testing on much larger samples (we only tested 20–30 children per training condition). If confirmed, these findings would provide some of the strongest empirical support to date for the role of prediction error in the acquisition of linguistic structure.

**Keywords:** priming; syntax; prediction; revision; learning





## 1. Introduction

As children develop their mastery of language, one of the skills they must gradually hone is understanding how words relate to one another and are combined to determine sentence-level meaning. As many of the papers in this special issue demonstrate, this knowledge covers not only the range of possible structures in the language, but also the likelihood of encountering different structures in the input. We have good evidence that children are sensitive to the frequency of particular structures (Arnon 2015) and that recent exposure to one type of structure (in either comprehension or production) increases the likelihood that such a structure will later be used by children (Messenger 2022). The reason as to why children show this sensitivity is less clear, however, as there are different theories regarding the underlying mechanisms (Pickering and Ferreira 2008). In this study, we test one potential mechanism, namely, error-driven learning.

Error-driven models of language acquisition (Chang et al. 2006; Dell and Chang 2014) propose that children's sensitivity to the frequency of different structures and their tendency to repeat a structure (i.e., structural priming) stem from the way they learn from linguistic input. Specifically, as children hear a sentence, they continuously generate expectations about each upcoming word. Such expectations are compared to the input (i.e., the actual next word), and, when there is a mismatch, a prediction error signal triggers the updating of long-term knowledge about linguistic structure. Thus, expectations are informed by previous linguistic experience and shape further learning.

To exemplify, in English, most transitive verbs are immediately followed by their object, which makes structures in which a different element intervenes between the verb and its

object relatively less frequent and, therefore, less expected. According to error-driven models, less frequent structures are more likely to be primed by recent experience, because their occurrence leads to larger prediction error signals, and these in turn cause greater changes to stored associative weights that determine which structure will be selected next (Chang et al. 2006; Jaeger and Snider 2013). This can explain, for example, why direct object (DO) datives—where the recipient of a transfer action is mentioned first after the verb and before the theme (object) of that action—may be easier to prime than prepositional object (PO) datives, where, following the general trend for English transitive verbs, the theme follows the verb and precedes the recipient (compare DO: *The girl will throw the turtle a ball* and PO: *The girl will throw a ball to the turtle*; recipient = *turtle*; theme = *ball*) (Rowland et al. 2012).

Moreover, violations of verb-specific structural preferences modulate the magnitude of both short-term (Peter et al. 2015) and cumulative priming effects (Fazekas et al. 2020) in production tasks, suggesting that children who are already able to produce both DOs and POs are sensitive to prediction errors generated by their knowledge of different verb biases for these structures. For example, stronger priming is observed for children following sentences such as *The zookeeper brings the giraffe some food* as compared to *The zookeeper gives the giraffe some food*. This is because in English child-directed speech, the DO sentence structure is much less likely to occur with the verb *bring* than *give* (Peter et al. 2015). Thus, stronger priming after a *bring* sentence than a *give* sentence is interpreted as evidence that children were more surprised by the occurrence of the DO with *bring*, which led to a further increase in the likelihood of re-using this structure.

However, no study has so far addressed whether sensitivity to prediction errors is evident in children's *understanding* of these structures, but it is important to demonstrate that children's acquisition of the less expected and harder-to-interpret DO structure is indeed driven by prediction error. In this study, we provide a direct test of this hypothesis and thus of the idea that prediction errors drive children's learning about the structure of their first language.

### 1.1. Why Are DOs Harder to Understand for Children?

The dative alternation is one of the best-studied structural alternations in English. Dative sentences describe the transfer of a theme to a recipient. As mentioned above, objects (patients and themes) tend to follow a transitive verb immediately in English sentences, resulting in a bias to expect the theme to be mentioned immediately after a dative verb (Thothathiri and Snedeker 2008). DO sentences violate this theme-first bias.

This may explain why children take longer to master DOs than POs (Osgood and Zehler 1981; Conwell and Demuth 2007; Arunachalam 2016), with many 4-year-olds still often mistakenly assigning the theme role to the first noun encountered after the verb when the sentence has a DO structure. Importantly, though, the fact that they violate an expectation may also provide a mechanism for the eventual acquisition of DOs: encountering the recipient where the theme was expected should generate a strong prediction error signal, leading to long-term changes to children's structural knowledge (to include the possibility that recipients may precede themes) and consequently to their structural expectations.

The violation of expectations can function as a learning mechanism in this way only if children can (i) reliably predict the theme and (ii) detect that the recipient has been mentioned in lieu of the theme. Can children do this? We know they have expectations about which types of entities are more likely to play the role of recipients vs. themes of a transfer event: specifically, they expect an inanimate entity (e.g., frisbee) to fill the theme role and an animate entity (e.g., duck) to fill the recipient role (Buckle et al. 2017)—we term this the animacy bias. We suggest that, provided the theme and recipient differ in animacy, children can use the animacy bias to both predict the (most likely) theme and detect whether the (most likely) recipient was mentioned immediately after the verb. We thus used the presence or absence of an animacy contrast in this study to manipulate the

strength of children's expectations and thereby test whether the acquisition of the DO structure is affected by prediction error.

*1.2. This Study: Manipulating the Strength of Children's Expectations*

During the training phase of our study, children listened to either 12 PO or 12 DO sentences. This manipulation of structure controlled for any improvement in comprehension scores due to structural priming effects (as there is little work looking at structural priming effects in comprehension tasks with children (Thothathiri and Snedeker 2008; Arai and Mazuka 2014; Bencini and Valian 2008). Our critical manipulation related to the visual displays accompanying the sentences (see Figure 1). These either showed one inanimate and one animate entity (e.g., frisbee and duck) or two animate entities (owl and duck). Sentences always described an agent (also depicted on the screen; e.g., Winnie the Pooh) transferring the theme (frisbee or owl) to the recipient (in this example, duck). Crucially, children who were looking at a frisbee and a duck could predict the frisbee would be the theme even before they heard the sentence; in contrast, children looking at an owl and a duck could not make this prediction. Thus, after hearing *Winnie the Pooh will give...*, children looking at displays containing an animacy contrast should generate a stronger expectation for the theme (predictable condition), compared to children looking at displays that do not feature an animacy contrast (unpredictable condition).

**Figure 1.** Schematic illustration of the experimental design, comprising pre-test (6 experimental trials, one example depicted), training (12 trials, one example in each of the four versions depicted), and post-test (6 experimental trials, one example depicted). The four between-participants training conditions were obtained by combining the two visual displays (predictable vs. unpredictable) and two sentence types (DO vs. PO). The displays are screenshots from the app (see below) and represent what participants actually saw in the experiment.

Note that these predictions should be generated (or not generated) in a similar way shortly after the verb regardless of whether children are listening to a sentence that will eventually unfold as a PO or a DO. But this predictability manipulation should more strongly or only affect learning for children trained on DOs: these children always hear the recipient first (e.g., *Winnie the Pooh will give...the duck...the frisbee/the owl*), resulting in a mismatch with expectations generated in the predictable condition, whereas children trained on POs always hear the theme before the recipient (e.g., *Winnie the Pooh will give...the*

*frisbee/the owl. . .to the duck*), with the result that their expectations are never disconfirmed in the predictable condition.

In sum, children trained on predictable DOs should first generate a strong prediction and then have it disconfirmed; children trained on predictable POs should also generate a strong prediction, which is not disconfirmed; and children trained on unpredictable DOs or POs should not generate a particularly strong prediction (so it should not matter as much whether this prediction happens to be confirmed or disconfirmed). In line with error-driven learning models (Chang et al. 2006), we hypothesized that stronger predictions that are later disconfirmed should be associated with greater prediction error and therefore greater learning, so we expected more learning after training on predictable DOs than unpredictable DOs (but no differences between predictable and unpredictable PO training conditions).

To measure learning, we asked children to listen to pre-recorded DO sentences while viewing pictures of the theme and recipient on a touchscreen and then act out their interpretation (e.g., dragging the theme picture towards the recipient picture). Children did this twice, in a pre-test before being exposed to the training sentences and in a post-test after; each test had a different set of six DO sentences. Three of the sentences in each set had animate recipients and animate themes (as shown in Figure 1), and the remaining three had animate recipients but inanimate themes (not shown in Figure 1), so that each child was tested on at least three screens that did not resemble the training screens they had been exposed to, and all children were tested on some animate-animate items. We expected the animinate-animate items to be more challenging, since animacy cues cannot aid their interpretation (i.e., these items can only be interpreted correctly if children know the DO structure).

We analyzed children's post-test comprehension scores (i.e., the percentage of sentences correctly acted out) as a function of the training condition, while controlling for pre-test comprehension scores as a continuous covariate. We expected children trained on predictable DOs to achieve higher comprehension scores than children trained on unpredictable DOs, and we expected no difference in performance between children trained on predictable and unpredictable POs.

## 2. Materials and Methods

### 2.1. Participants

We recruited 151 English-learning children between the ages of 48 and 59 months (i.e., 4-year-olds). All testing took place in person between April 2019 and August 2022 (with an interruption during the COVID-19 pandemic). Fifty-three of the children were tested in a quiet lab at Cardiff University Centre for Human Developmental Science (UK); the remaining children were tested at a number of different nurseries in Cardiff and South Wales. We did not collect socio-demographic or ethnicity information, but almost all children were white, and a substantial majority were from lower to upper middle-class backgrounds, with only a minority from working-class backgrounds. Data from 24 children (all in the unpredictable DO training condition) were discarded after it was discovered that one of the training sentences in this condition was mistakenly presented in the prepositional object (instead of the direct object) form (the audio file was then replaced with the correct version). A further 7 children were discarded because of technical glitches (6, poor wi-fi connection, which meant the data were not uploaded correctly or the touchscreen task froze mid-way through the session) or because their caregivers disclosed that they had received a diagnosis indicating their language development may be atypical (1, ASD).

The final sample was comprised of 120 English-learning children, 56 male, with an average age of 53.30 months (SD = 4.36). Due to recruitment constraints, 3 of the children were tested in the month before they turned 4, and 8 of the children had already turned 5 by the time testing took place (see Table 1 for a breakdown of age by training condition). Two children were reported to be dominant in a language other than English at the time of testing, 5 children were acquiring English and one other language in roughly balanced proportions, and 12 children were receiving some input in a language other than English

but were reported to be dominant in English. As these children all lived in Wales, and the majority had started attending school (where Welsh is a compulsory subject), they were all exposed to Welsh to some extent in addition to English, but the figures above only include those children with a significant amount of Welsh exposure either at home or at school (i.e., attending a Welsh-medium school).

**Table 1.** Sample size and age by training condition.

| Training | N | Mean Age (Months) | Age Range |
|---|---|---|---|
| DO Unpred | 25 | 52.24 | (48–63) |
| DO Pred | 28 | 51.96 | (47–63) |
| PO Unpred | 32 | 54.84 | (48–67) |
| PO Pred | 35 | 53.71 | (47–68) |

Written consent was obtained from a caregiver for all participants, and children verbally assented to take part in the study. The study received ethical approval from the Ethics Committee of the School of Psychology, Cardiff University (approval number: EC.18.05.08.5295GR).

Power and Sample Size

We did not conduct an a priori power analysis. After collecting a first wave of data (N = 97, of which 20 were included in the DO unpredictable and 22 in the DO predictable training conditions), we conducted some power simulations using the R package simR (https://osf.io/2c9tx/, URL accessed on 1 September 2023), which suggested that 35 participants per between-subject training condition should achieve 80% power given an effect size estimate of d = 0.82 (observed effect size) for the comparison between the DO predictable and DO unpredictable training conditions. Due to recruitment constraints, we were unable to reach this sample size (which might itself have been an underestimation, since the observed effect size is likely inflated; Albers and Lakens 2018). We report this here to help inform interpretations of the findings below, as it is likely this study was underpowered. Note that this power analysis only used data from those pre- and post-sentences that had animate recipient and animate themes (for reasons detailed below).

*2.2. Materials and Procedure*

Using a novel web-based touchscreen task (demo: https://languagetasks.warwick.ac.uk/structure-pred-example/; code: https://osf.io/2c9tx/, URLs accessed 1 September 2023), we first assessed children's baseline comprehension of DO sentences (pre-test, see Figure 1), then exposed them to one of four different training conditions (between-participants), and finally, we assessed their DO comprehension skills again (post-test), using a different set of DO sentences. A full list of materials is available at https://osf.io/2c9tx/ (URL accessed 1 September 2023).

2.2.1. Pre-Test and Post-Test

At the start of the session, children practiced dragging and dropping the characters that would later appear as agents of the pre-test sentences. This allowed the experimenter to check whether the children understood the characters' names (which was the case almost universally) and—if they did not—to familiarize them with the names, so that they would find it easier to later process the start of the sentences. After this brief familiarization phase, children listened to 10 pre-recorded sentences while viewing sets of three pictures on the tablet screen. One picture (always displayed centered in the top half of the screen) depicted the agent/subject of the sentence, whereas the other two pictures (always displayed on the bottom half of the screen, right-left order randomized on each trial) displayed the theme/object and a third entity. In test dative sentences (6), this third entity was the recipient of the DO sentences (always animate), whereas in filler sentences (4), this could be a location (e.g., Now, the boy will put the train in the box) or instrument (e.g., In this

one, Boots will clean the chair with the sponge). Each sentence was followed by a question asking children to act out the meaning of the sentence (e.g., Can you put the train in the box?). Correct actions required dragging the theme/object towards the recipient (for DO test sentences) or the location (for some filler sentences) or using the instrument to perform an action on the theme/object (for other filler sentences). Children repeated the same task, including the familiarization phase, after the training phase (see below). This post-test included 10 new sentences (again, 6 DO sentences and 4 new fillers of the same types as those encountered in the pre-test). Since the assignment of sentences to the pre vs. post-test was counterbalanced across participants, we describe below how all of these materials (12 DO sentences and 8 filler sentences) were created.

The dative verbs used in DO sentences were *bring*, *throw*, and *send*. Each verb was used twice in the pre-test and twice in the post-test, once in a DO sentence with an inanimate theme and an animate recipient and once in a DO sentence with animate theme and recipient. The rationale for choosing these particular verbs was as follows. *Send*, *bring*, and *throw* are PO-biased verbs in English (according to Peter et al. 2015, who analyzed the Manchester Corpus from CHILDES, MacWhinney 2000), meaning they are less likely to appear in DO constructions. This made the task harder for children, avoiding ceiling effects. Note that pairing verbs with their least preferred structure may have also had another consequence, namely, that the pre-test and post-test sentences may have generated prediction error signals that also contributed to children learning over and above any effects of the training sentences (note, though, that during pre- and post-tests, children received no explicit feedback on whether their interpretation was correct, unlike in training). Importantly, these effects would have been the same for all children, regardless of training condition. Filler sentences used the following verbs: *bounce* (1 sentence), *put* (2 sentences), *move* (2), *poke* (1), *tickle* (1), *clean* (1).

The following characters were used as agents/subjects of the test and filler sentences (with some repeated more than once): *boy*, *girl*, *Peppa Pig*, *king*, *queen*, *Emily Elephant*, *Boots*, *princess*, *prince*, *Dora the explorer*. Aside from the agent/subject, other verb arguments in filler sentences were all inanimate (*ball*, *bat*, *train*, *box*, *hat*, *table*, *jelly*, *fork*, *teddy bear*, *feather*, *chair*, *sponger*, *food*, *basket*, *blanket*, *sofa*).

The following nouns were used as inanimate themes in test DO sentences: *cake*, *clock*, *toy*, *present*, *ice-cream*, *balloon*. Animate themes and recipients were *giraffe*, *elephant*, *hamster*, *turtle*, *squirrel*, *monkey*, *horse*, *mouse*, *rabbit*, *tiger*, *lion*, *crocodile*, *cat*, *puppy*. Some of the animate nouns were used more than once, because we struggled to find a sufficient number of items that were age-appropriate and easily depictable. When words were repeated, we made sure to counterbalance the role (theme/recipient) they played in the sentences and whether they played the same or a different role on their second occurrence. This means that even numbers of repeated words played the same role across repetitions or switched roles (and if the latter, theme → recipient or recipient → theme switches were equally likely). All selected common nouns had test-based age-of-acquisition norms of 2 (except *sofa*, which had a value of 4), and age-of-acquisition norms based on adult ratings were below 6 for all nouns (both sets of norms from Brysbaert and Biemiller 2017). We also took care to match the average age of acquisition (based on adult ratings) across the two sets of sentences used in the pre- vs. post-test. In addition, words that appeared together in the same DO sentence as theme and recipient never shared the same onset phoneme. As much as possible, we tried to avoid pairing together words that had strong associative relationships (e.g., *cat* and *dog*) and/or shared visual features (e.g., *hamster* and *mouse*).

The DO sentences followed this pattern: Preamble + the + Agent + will + verb + the + Recipient + a/an + Theme. We chose to use the indefinite determiner before the theme after piloting the study on a 4-year-old and a 6-year-old child and finding that their comprehension performance was lower than expected. Previous studies have shown that 4-year-olds are able to interpret at least some double object (DO) sentences even when these use novel verbs (Rowland and Noble 2010; Rowland et al. 2014), but our pilot participants struggled to understand DO sentences at all when the theme and recipient were both

animate, e.g., *In this one, the boy will throw the turtle the cat*. Rowland and Noble (2010) found that using a noun phrase for the theme and a proper noun for the recipient helped children to differentiate between the theme and recipient (e.g., *In this one, the boy will throw Turtle the cat*). However, this option was not pragmatically appropriate for our stimuli (as the pictures were all photographic images of real animals and objects, rather than cartoon depictions). Thus, to help children differentiate the theme from the recipient, we used the indefinite determiner for the former and the definite determiner for the latter (e.g., *The boy will throw the turtle a cat*); this sounded more pragmatically felicitous than the alternative (i.e., *The boy will throw a turtle the cat*). All sentences (including the ones used during training, see below) were recorded by the same female native speakers of English using child-directed prosody. The speaker was instructed to speak slowly, clearly demarcating out each word and putting emphasis on the main verb, the theme, and the recipient regardless of sentence structure.

2.2.2. Training

All children completed a training session between the pre-test and post-test sessions, during which they were exposed to 12 dative sentences. The experimenter modeled the correct interpretation of each sentence, and the child was instructed to mimic their action. Between children, we varied whether the dative sentences were all POs or DOs as well as whether they were predictable (inanimate theme) or unpredictable (animate theme), as described in the introduction. The crossing of these two manipulations resulted in 4 between-participants training conditions. An example item in the four conditions is shown in Figure 1.

Unlike pre-/post-test sentences, training sentences used only definite noun phrases, because we were not concerned about children interpreting these sentences incorrectly; in fact, we wanted to encourage such incorrect interpretations (i.e., interpreting the first post-verbal noun of a DO sentence as the theme) to generate prediction error in relevant conditions. We took care to avoid lexical overlap between pre-/post-test materials and training materials, but due to the limited number of ditransitive verbs that are known by 4-year-olds, *bring, send,* and *throw* were also used as verbs in the training sentences (2 each); in addition, *give, show,* and *pass* were each used in 2 training sentences. The following characters were used as agents: *boy, girl, Tigger, Rebecca Rabbit, George, Piglet, Bob the Builder, Chase, Dora the Explorer, Boots, Marshall, Winnie the Pooh*. Inanimate themes (in the predictable conditions) were *cheese, bottle, pencil, brush, dress, plate, shoe, biscuit, spoon, flower, gloves, frisbee*. Animate themes (in the unpredictable conditions) were *dinosaur, butterfly, frog, fox, bird, ant, dog, deer, cow, pig, goat, owl*. Animate recipients (in all conditions) were *bear, zebra, chicken, donkey, sheep, fish, penguin, kitten, hippo, caterpillar, bee, duck*. All selected common nouns had test-based age-of-acquisition norms of 2 (except *caterpillar, biscuit*: 4; *frisbee:* 6), and age-of-acquisition norms based on adult ratings were below 6 for all nouns (Brysbaert and Biemiller 2017). The average age of acquisition (based on adult ratings) was matched across predictable and unpredictable training conditions to ensure children would not find one set of sentences more difficult than the other. Theme and recipient always started with different phonemes and were as semantically unrelated as possible.

Before starting the session, children practiced dragging and dropping the characters that would later appear as agents of the training sentences. Then, the experimenter informed the child that they would switch roles and that the experimenter would play the game on the tablet, while the child would copy the experimenter's actions using laminated cards depicting the theme and recipient exactly as they appeared on the tablet screen. At the start of each trial, before playing the sentence, the experimenter placed the cards on the table underneath the corresponding pictures on the tablet. They then triggered playback of the sentence and showed the child the correct act-out for that sentence on the tablet, encouraging the child to copy their action using the cards. Since it was important to ensure that the children paid attention to the correct interpretation of the sentences during training, the experimenter was instructed to repeat the action on the tablet and/or model the action with the cards if needed (e.g., if the child was hesitant or copied the action incorrectly).

Sessions were not recorded, but the experimenter's response on the tablet was recorded, allowing us to check for experimenter errors.

*2.3. Data Analysis*

The key analysis looked at accuracy on experimental (DO) trials in the post-test as a function of (a) accuracy on experimental (DO) trials in the pre-test (centered; to control for baseline differences in ability between children), (b) sentence type (DO vs. PO, reference level: PO), (c) predictability (Pred vs. NonPred, reference level: NonPred), (d) trial difficulty (animate-animate = hard vs. animate-inanimate = easy, reference level: animate-inanimate). Recall that (b) and (c) relate to the training condition to which children were assigned and are thus between-participant factors; instead, (d) relates to the type of test trial and is a within-participant factor. Importantly, note that although both (c) and (d) refer to a manipulation of animacy of the theme, they are treated as separate factors in the analysis, because the predictability manipulation (c) was implemented in the training phase, whereas the trial difficulty manipulation (d) was implemented in the pre-test and post-test phases. All categorical predictors were contrast coded ($-0.5$, $0.5$) and centered, and the model also included all possible interactions between the three categorical predictors (see https://osf.io/2c9tx/, analyses_with_power folder, html file, Section 4, for the full R syntax; URL accessed 1 September 2023). The model was fit using the *glmer* function from the lme4 package (Bates et al. 2023) in R (Version 3.4.1, R Development Core Team n.d.). Ninety-five percent confidence intervals were computed with the *confint* function (method = "Wald"). The model only included random intercepts for participants; random slopes by participants for trial difficulty and pre-test accuracy and random intercepts and slopes by items could not be estimated. Note that items were defined according to the verb (bring, throw, show), so there were only 3 items in the study. This likely accounts for difficulties in estimating by-item random variation.

**3. Results**

We first checked for experimenter errors in the training phase: These were rare, accounting for less than 10% of the training trials in each training condition (DO NonPred: 100% accuracy, DO Pred: 96%, PO NonPred: 93%, PO Pred: 97%). Note that not all of these were instances of genuine experimenter errors, as children were sometimes reluctant to stop using the tablet, and we allowed them to continue providing their response on the tablet while the experimenter modeled the correct action for them using the cards. We then looked at children's performance on filler trials in the pre- and post-test. Accuracy was above 80% in both phases for children in all training conditions (see https://osf.io/2c9tx/, URL accessed 1 September 2023, analyses_with_power folder, html file, Section 2, for a full breakdown) and was numerically very similar across the pre- and post-test, indicating that children were generally attentive, understood the task, and did not become excessively fatigued at the end of the study.

Table 2 reports average percentage accuracy in the pre-test and post-test as a function of training condition, separately for experimental items that were expected to be easier (animate-inanimate DOs, AI) or harder (animate-animate DOs, AA). Figure 2 reports difference scores (i.e., post-test accuracy—pre-test accuracy) as a function of the training condition, separately for experimental items that were expected to be easier (AI) or harder (AA).

*3.1. Planned Analyses*

There were four main findings (see Table 2 and Figure 2). First, children's performances generally improved from the pre-test (average: 41.94%, SD = 36.59%) to the post-test (average: 53.19%, SD = 36.83%). Note that we did not test for this statistically, but the numerical trend can be observed in Table 2. Though this suggests that children overall became better at the task with practice, the average accuracy was still far from ceiling during the post-test. During the pre-test, performance was actually lower than chance (50%,

between-participant one-sample *t*-test: $t(119) = -2.75$, $p = 0.007$), suggesting that children's interpretations may have been driven by a systematic bias to interpret the first post-verbal NP as the theme. These trends were underscored by substantial interindividual variability, as indicated by the very large standard deviations in Table 2 and the wide confidence intervals in Figure 2.

**Table 2.** Pre-test and post-test accuracy (%). By-subject means and standard deviations.

| Sentence Type | Predictability | Trial Difficulty | Mean Accuracy (SD) | |
|---|---|---|---|---|
| | | | Pre-Test | Post-Test |
| DO | NonPred | AA | 41.33 (33.72) | 41.33 (37.61) |
| DO | NonPred | AI | 48 (39.77) | 54.67 (39.53) |
| DO | Pred | AA | 36.9 (34.35) | 58.33 (28.15) |
| DO | Pred | AI | 55.95 (34.01) | 69.05 (28.59) |
| PO | NonPred | AA | 34.38 (35.4) | 40.62 (34.64) |
| PO | NonPred | AI | 47.92 (40.55) | 59.38 (39.47) |
| PO | Pred | AA | 33.33 (35.24) | 39.05 (35.69) |
| PO | Pred | AI | 40.95 (37.12) | 63.81 (39.08) |

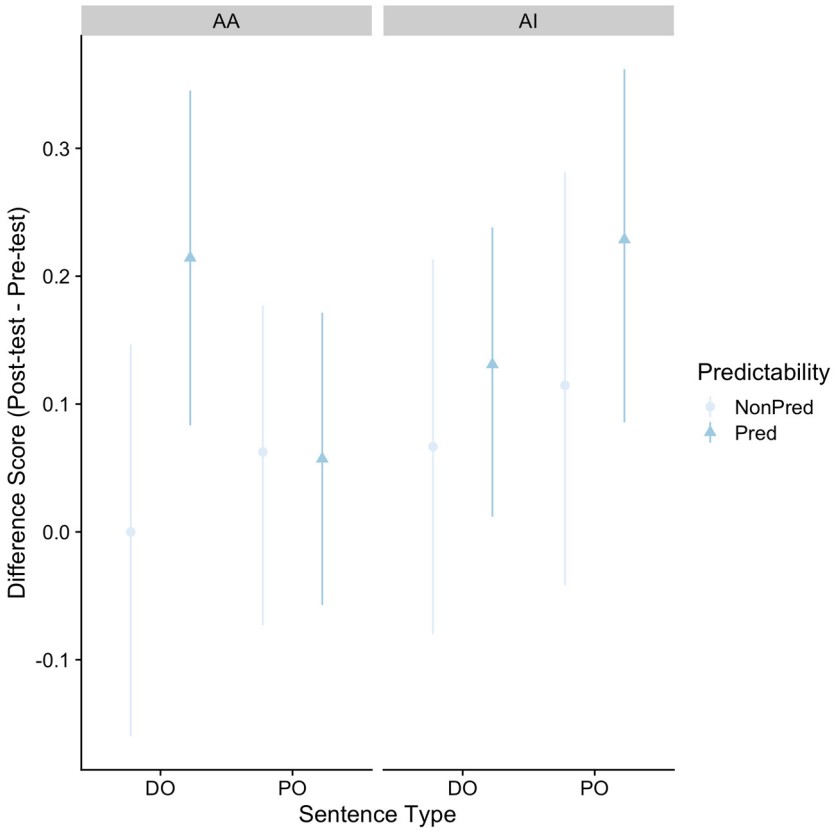

**Figure 2.** Difference scores. Point estimates represent means, and the error bars represent 95% bootstrap confidence intervals. Left panel represents more difficult animate-animate test trials, and right panel is based on less difficult animate-inanimate test trials.

In addition, and as expected, children were more accurate when acting out the easier animate-inanimate (average across pre- and post-tests: 54.86%, SD = 38.02%) than the animate-animate items (average pre-/post-test: 40.28%, SD = 34.74%), which confirms that children can exploit animacy cues when interpretating dative sentences (Buckle et al. 2017). However, and most importantly, the interaction between sentence type and predictability was not significant ($p = 0.255$), nor were the main effects of sentence type (i.e., no priming; $p = 0.451$) or predictability ($p = 0.149$). Recall that we were expecting the interaction to

be significant, as children trained on predictable DOs were expected to show larger improvements in comprehension accuracy than those trained on unpredictable DOs, whereas we were expecting no difference for children trained on predictable vs. unpredictable POs. The three-way interaction between sentence type, trial difficulty, and predictability was also not significant ($p = 0.373$). Finally, and unsurprisingly, pre-test accuracy was a significant predictor of post-test accuracy ($p = 0.005$). Full model output is available at https://osf.io/2c9tx/, URL accessed on 1 September 2023, analyses_with_power folder, html file, Section 4.1 (Omnibus ANCOVA).

### 3.2. Additional Exploratory Analyses

The analyses above did not reveal any effect of training condition. In additional analyses, we decided to explore further the interaction between sentence type and trial difficulty ($p = 0.092$). Figure 2 displays difference scores (accuracy in post-test minus accuracy in pre-test) for the four training conditions (two sentence type by two predictability conditions), separately for the easier AI test items (right panel) and the harder AA test items (left panel). Looking at this figure, children trained on POs tended to show somewhat larger increases in performance on the easier animate-inanimate trials compared to the harder animate-animate trials (17.41% vs. 5.97%), regardless of predictability. In contrast, children trained on DOs showed (if anything) slightly larger performance gains on the harder animate-animate trials (11.32% vs. 10.06% on AI test trials). Though we had not expected animate-animate and animate-inanimate trials to show different effects of the training manipulations, in an attempt to better understand our findings, we looked at DO-trained groups and PO-trained groups separately (see https://osf.io/2c9tx/, URL accessed on 1 September 2023, analyses_with_power folder, html file, Section 4.3), modeling the post-test data in each group as a function of pre-test performance and the interaction between predictability and trial difficulty. In both groups, we observed that accuracy was overall higher for the easier animate-inanimate than the harder animate-animate test items (DO-trained, $p = 0.03$, PO-trained, $p < 0.001$), though this effect was numerically larger in PO-trained groups. Unsurprisingly, higher pre-test accuracy tended to predict better post-test accuracy (DO-trained, $p = 0.03$, PO-trained, $p = 0.062$). In PO-trained groups, neither predictability ($p = 0.752$) nor the interaction between predictability and trial difficulty were significant ($p = 0.406$). In DO-trained groups, children trained on predictable DOs tended to show higher accuracy in the post-test (controlling for pre-test performance) compared to children trained on unpredictable DOs (effect of predictability, $p = 0.051$); the interaction between predictability and trial difficulty was once again not significant ($p = 0.665$), suggesting that any beneficial effect of DO predictable training (to the extent that it was present) transferred similarly to AA and AI test items. Thus, when comparing children trained on predictable and unpredictable DOs, there was a numerical trend in the direction predicted by error-driven learning models (i.e., more learning following training on sentences that generate larger prediction errors).

### 4. Discussion

In this study, we set out to test the role of error-based learning in 4-year-olds' acquisition of the difficult DO sentence structure. We employed a pre-/post-test design with children assigned to one of four training groups, orthogonally manipulating the type of sentence structure they were exposed to and whether the sentences (in combination with the visual context) allowed them to make strong predictions about the identity of the theme or not. We hypothesized that when children's strong predictions were disconfirmed (i.e., in the predictable DO training conditions), children would see the most benefit to their learning compared to conditions in which they either could not generate predictions (unpredictable DO/PO), or they could generate strong predictions, but these were not disconfirmed (predictable PO). Uniquely, we tested whether encouraging children to generate predictions that were later disconfirmed improved their ability to accurately interpret both easier and harder (i.e., with or without animacy cues) DO sentences.

We found some—very preliminary—indication that children's comprehension abilities might benefit more when they were exposed to predictable DOs than unpredictable DOs (exploratory analyses). In contrast, predictability did not make a difference for children trained on POs. These findings are compatible with error-based learning approaches to the acquisition of syntactic structure (Chang et al. 2006). We argue that—compared to previous evidence cited in support of these approaches—our study makes two significant novel contributions.

First, previous work (Peter et al. 2015; Fazekas et al. 2020) has demonstrated an effect of error-based learning in production tasks, whereas this is the first study to investigate whether such an effect occurs in comprehension. This is important, because it is generally assumed that children's comprehension abilities predate the development of their production abilities. Therefore, showing that prediction error affects comprehension is crucial to support the claim that this mechanism can support learning of syntactic structure from the very earliest stages of development (of course, convincingly showing this would require testing much younger children than those in our study).

Second, previous demonstrations of error-driven learning of structure have focused on children's sensitivity to verb-related structural biases (Peter et al. 2015; Fazekas et al. 2020). This work assumes that children have already learned to expect either a PO or DO, depending on the lexical item that functions as the main verb in the sentence. In contrast, the mechanism we tested would allow children to learn about the DO in the first place based simply on generating context-specific lexical expectations. Such expectations—we assume—would be based on combining event-based knowledge about which kind of entities are more likely to be themes with structural knowledge (i.e., themes tend to be inanimate objects and mentioned right after the verb); see Gambi et al. (2016) for a demonstration that 3- to 5-year-olds are able to generate and combine these two types of expectations. Thus, we believe this mechanism could explain how children learn about the possibility that recipients might be mentioned before themes in the first place.

However, it is important to acknowledge that these findings are preliminary and need to be confirmed in future research before any strong implications for theory can be drawn. The main limitation is related to the small number of test items (6), which was compounded by the unexpected finding that the effect of the training condition seemed to differ for the difficult animate-animate DOs compared to the easier animate-inanimate DOs. We have since been conducting a follow-up conceptual replication of this study in which all test items are of the difficult animate-animate kind. Though increasing the overall number of items is challenging when testing children of this age, it is well known that power depends on the number of items as well as on the number of participants in crossed random effects designs (Westfall et al. 2014). In addition, we observed very large individual differences in children's baseline performance. However, since we randomly assigned children to training conditions, we were only able to control for these post-hoc, by including pre-test performance into our models. A better approach would involve pre-testing children on a different measure of grammatical knowledge (e.g., Test for the Reception of Grammar, Bishop 2003) and then assigning children to conditions based on their scores to ensure roughly matched samples. We have implemented this approach in our follow-up study. Finally, it should be noted that, like the studies on verb bias mentioned above (Peter et al. 2015; Fazekas et al. 2020), we also assumed that children would generate certain expectations based on our experimental manipulations, but we did not actually measure whether (or the extent to which) they did so. Given we know there are large individual differences in expectation generation that relate to children's linguistic knowledge (Borovsky et al. 2012; Mani and Huettig 2012; Gambi et al. 2021), it is reasonable to expect that not all children will generate equally strong expectations and may therefore benefit from error-based learning to differing extents. For this reason, in our follow-up study, we are recording children's eye movements in real time to measure their prediction abilities.

One final limitation of our study is the fact that we only included an immediate post-test. In order to demonstrate learning effects, future work should consider including delayed post-tests to examine whether any improvements in accuracy persist over time and—if so—for how long. However, one encouraging finding was that we observed the (numerically) largest improvements in performance for difficult animate-animate DOs following training on easier animate-inanimate DOs. This suggests that children were not simply getting better at the task at hand, but they were to some extent able to generalize what they learned from one type of DO sentence to another.

Before concluding, two unexpected aspects of our findings merit discussion. We included both PO and DO training conditions to control for structural priming effects, but we in fact found no evidence that children benefitted more overall from DO than PO training conditions. Though there is evidence for priming of online interpretation in comprehension (Thothathiri and Snedeker 2008), even in children, here, we collected only off-line act-out responses. Indeed, previous work has shown that comprehension priming is highly dependent on the type of measure used, and it is difficult to observe in the absence of lexical overlap (Tooley and Traxler 2010; Tooley et al. 2019). Moreover, children were exposed to multiple DOs in the pre-test and post-test, and it is possible that this blocked design masked any priming effect from the training phase.

By far, the strongest effects in our study were observed when comparing DOs whose interpretation was aided by animacy cues and DOs where animacy cues could not help children identify the to-be-moved theme. Though we expected children to make use of animacy cues when present, we did not expect these would have a larger effect for children trained on POs than children trained on DOs. One possibility is that PO training reinforced children's expectations for the to-be-moved object to be the one mentioned immediately after the verb. This meant children's performance on the difficult animate-animate test items did not improve from pre-test to post-test, but children were still able to use the animacy cues to interpret the easier animate-inanimate test items, and they likely got better at doing so as they gained more practice at the task. In contrast, DO training, and particularly DO predictable training, may have caused children to weaken their expectation for the to-be-moved object (theme) to be mentioned immediately after the verb, which meant they would have strengthened weaker structural knowledge (i.e., recipients are sometimes mentioned immediately after the verb) that aided their interpretation of the difficult animate-animate items, effectively reducing the accuracy difference between animate-animate and animate-inanimate test items.

In sum, we found that children exposed to sentences that encouraged the generation of incorrect linguistic predictions improved their comprehension numerically more than those who were exposed to sentences that did not support predictions. These preliminary findings need to be confirmed by additional testing on much larger samples. If confirmed, they would provide some of the strongest empirical support to date for the role of prediction error in the acquisition of linguistic structure.

**Author Contributions:** Conceptualization, C.G. and K.M.; methodology, C.G. and K.M.; software, C.G.; investigation, C.G.; resources, C.G.; data curation, C.G.; writing—original draft preparation, C.G.; writing—review and editing, C.G. and K.M.; visualization, C.G. supervision, C.G.; project administration, C.G. and K.M.; funding acquisition, C.G. and K.M. All authors have read and agreed to the published version of the manuscript.

**Funding:** This research was funded by a British Academy/Leverhulme Small Research Grant, grant number SRG1920_100600 to C.G. and K.M.

**Institutional Review Board Statement:** The study was conducted in accordance with the Declaration of Helsinki and approved by the Ethics Committee of omitted University, School of Psychology omitted. (protocol code EC.18.05.08.5295GR, date of approval: 18 May 2018).

**Informed Consent Statement:** Written informed consent was obtained from a relevant caregiver for all children involved in the study; in addition, all children provided verbal assent to the study.

**Data Availability Statement:** Data for this study can be found at https://osf.io/2c9tx/ (accessed on 1 September 2023).

**Acknowledgments:** Several amazing research assistants helped prepare the materials for this study and collected the data: Zoe Baberg-Collins (neè Williams), Lucie Smith, Brianna Bowen, Charlotte Draper, Vidushi Agarwal, Chara Sofocleous, Aleksandra Dummer. We also want to thank all the children and families who donated their time, and the managers and staff at participating nurseries.

**Conflicts of Interest:** The authors declare no conflict of interest. The funders had no role in the design of the study; in the collection, analyses, or interpretation of data; in the writing of the manuscript; or in the decision to publish the results.

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
