# Peer review of "The Role of Prediction Error in 4-Year-Olds’ Learning of English Direct Object Datives"

_languages, doi:10.3390/languages8040276_

Round 1
Reviewer 1 Report
Comments and Suggestions for Authors
This paper reports a study which aims to test the idea that the development of understanding of the Double Object (DO) dative is driven by the child’s sensitivity to prediction error. The paper is well-written and the study it reports is well-designed, though, as the authors themselves note, it is probably underpowered to detect the critical interaction. However, the authors do find some preliminary support for their hypothesis in the form of a marginal (p=.051) effect of predictability on comprehension of the DO dative in an exploratory analysis.
I think the authors do a good job of walking the line between acknowledging that their pre-registered analyses failed to provide support for their hypothesis and throwing the baby out with the bath water – and agree that the marginal effect that they report is potentially important, though, as they themselves note, this finding is very preliminary and needs to be confirmed in future research.
Minor Point
The authors undersell their study slightly be presenting it as an attempt to ‘fill a gap in the literature’ (lines 64-65). Filling a gap in the literature is not a very strong rationale for a study. Better to motivate the study in terms of what it will add/why it is important than in terms of filling a gap in the literature (which may not be worth filling).
Author Response
Reviewer 1
This paper reports a study which aims to test the idea that the development of understanding of the Double Object (DO) dative is driven by the child’s sensitivity to prediction error. The paper is well-written and the study it reports is well-designed, though, as the authors themselves note, it is probably underpowered to detect the critical interaction. However, the authors do find some preliminary support for their hypothesis in the form of a marginal (p=.051) effect of predictability on comprehension of the DO dative in an exploratory analysis.
I think the authors do a good job of walking the line between acknowledging that their pre-registered analyses failed to provide support for their hypothesis and throwing the baby out with the bath water – and agree that the marginal effect that they report is potentially important, though, as they themselves note, this finding is very preliminary and needs to be confirmed in future research.
Many thanks for your positive appraisal of our work. We did indeed try very hard to be cautious in our interpretation and we are pleased that came through.
Minor Point
The authors undersell their study slightly be presenting it as an attempt to ‘fill a gap in the literature’ (lines 64-65). Filling a gap in the literature is not a very strong rationale for a study. Better to motivate the study in terms of what it will add/why it is important than in terms of filling a gap in the literature (which may not be worth filling).
Thanks for this point, we appreciate it. We have rephrased the relevant paragraph as follows to better convey the importance of the study (lines 64-68).
“However, no study has so far addressed whether sensitivity to prediction errors is evident in children’s understanding of these structures, but it is important to demonstrate that children’s acquisition of the less expected and harder-to-interpret DO structure is indeed driven by prediction error. In this study, we provide a direct test of this hypothesis and thus of the idea that prediction errors drive children’s learning about the structure of their first language.”
Reviewer 2 Report
Comments and Suggestions for Authors
This comprehension study tested whether children’s erroneous predictions of sentence structure for dative sentences, based on animacy cues, could lead to learning of the DO dative structure, in line with error-based learning theories. The authors argue that the study provides preliminary evidence in support of this, although it is under-powered, and therefore findings are not significant. Results could therefore be taken to support the findings from previous studies investigating production (vs. comprehension).
The article provides a very clear rationale, based on previous literature, and the concept of error-based learning is well explained. The methods are fairly complex, but a comprehensive description is given. Results are reported clearly and concisely, and are easy to follow. The Discussion is clear, with useful suggestions for future studies. Overall, I found this an enjoyable paper to read, despite the sometimes complex terminology and (necessary) lengthy description of the method.
I do have a few reservations, though, and suggestions for improvements, before the paper is accepted for publication.
· While the hypothesis is given (lines 134-6), I would have found it helpful for the specific conditions to be laid out here in terms of where higher rates of improvements would be expected. This is simply due to the complexity of terminology and how much there is to take in, particularly for those less familiar with the dative alternation.
· It seems particularly problematic that some of the children were only just acquiring English (line 176); others having Welsh as their dominant language also seems problematic. English and Welsh have rather different syntactic structures (I’m not sure it has the equivalent of a DO?), so there is the possibility of transfer effects. Even if this weren’t the case, though, having less knowledge of English may affect the children’s performance, e.g., limit the effects of the experimental manipulation. These children should therefore be excluded from the analysis, unless the authors are able to clearly justify why this is not a problem. It might even help with the non-significant results/high SDs.
· Why was a power analysis only conducted part way through? This raises concerns for me, e.g., whether the analyses were run with just this initial wave of data collection, before more data were collected when results were not significant. However, given the lack of statistically significant findings in the final sample, it seems irrelevant to ask for any earlier results to be reported (although they could be added to OSF, of course, for transparency, if they exist).
· I’m not really sure what the following sentence is saying: “When words were repeated, we made sure to counterbalance which role (theme/recipient) they played in the sentences and whether they played the same or a different role on their second occurrence.” (lines 257-9) Did these repeated nouns always appear in different roles? It’s not clear to me.
· In frequentist statistics, results are either significant or not. Please therefore rephrase “marginal interaction” (line 394) and “a marginal effect of predictability” (lines 411-12), e.g., to describe these as trends. Similarly, lines 437-8 imply that an effect was found, whereas the stats don’t support this. Have the authors considered running Bayesian analyses instead? This would likely be of benefit in terms of the conclusions that can be drawn.
Author Response
Reviewer 2
This comprehension study tested whether children’s erroneous predictions of sentence structure for dative sentences, based on animacy cues, could lead to learning of the DO dative structure, in line with error-based learning theories. The authors argue that the study provides preliminary evidence in support of this, although it is under-powered, and therefore findings are not significant. Results could therefore be taken to support the findings from previous studies investigating production (vs. comprehension).
The article provides a very clear rationale, based on previous literature, and the concept of error-based learning is well explained. The methods are fairly complex, but a comprehensive description is given. Results are reported clearly and concisely, and are easy to follow. The Discussion is clear, with useful suggestions for future studies. Overall, I found this an enjoyable paper to read, despite the sometimes complex terminology and (necessary) lengthy description of the method.
Many thanks for these positive comments on our work.
I do have a few reservations, though, and suggestions for improvements, before the paper is accepted for publication.
- While the hypothesis is given (lines 134-6), I would have found it helpful for the specific conditions to be laid out here in terms of where higher rates of improvements would be expected. This is simply due to the complexity of terminology and how much there is to take in, particularly for those less familiar with the dative alternation.
Thank you for this helpful suggestion. To improve clarity we have specified earlier on and close to where we first state the hypothesis which conditions were expected to lead to higher rates of improvement (textual additions in bold in the passage below, lines 136-138).
“In sum, children trained on predictable DOs should first generate a strong prediction and then have it disconfirmed; children trained on predictable POs should also generate a strong prediction which is not disconfirmed, while children trained on unpredictable DOs or POs should not generate a particularly strong prediction (so it shouldn’t matter as much whether this prediction happens to be confirmed or disconfirmed). In line with error-driven learning models [4], we hypothesized that stronger predictions which are later disconfirmed should be associated with greater prediction error and therefore greater learning, so we expected more learning after training on predictable DOs than unpredictable DOs (but no differences between predictable and unpredictable PO training conditions).”
- It seems particularly problematic that some of the children were only just acquiring English (line 176); others having Welsh as their dominant language also seems problematic. English and Welsh have rather different syntactic structures (I’m not sure it has the equivalent of a DO?), so there is the possibility of transfer effects. Even if this weren’t the case, though, having less knowledge of English may affect the children’s performance, e.g., limit the effects of the experimental manipulation. These children should therefore be excluded from the analysis, unless the authors are able to clearly justify why this is not a problem. It might even help with the non-significant results/high SDs.
Thanks for allowing us to clarify this point. None of the children included in the analysis were “only just acquiring English” at the time they were tested and the vast majority were dominant in English. All the children were exposed to English in the home environment and/or at nursery/school. As stated in the manuscript, not all the children were being raised monolingually and – in fact – given they all resided in Wales, all of them had some varying level of exposure to the Welsh language. Importantly however, only two of the 120 children were reported as having more exposure to a language other than English than to English at the time of testing (this was never Welsh, by the way: it was Portuguese for one child and Flemish Dutch for the other); an additional 17 children were exposed to a language other than English either just as much as or less than English. As far as we are aware Welsh does not have the equivalent of a DO construction (“Rhys gives Angharad the book” translates to a PO “Mae Rhys’n rhoi y llyfr i Anghard”), however since no child was dominant in Welsh this should not raise transfer issues. Since – as noted – the study is likely already underpowered and should be replicated, we judged that it would be best to include all available participants even if this may come at the cost of higher between participant variability.
- Why was a power analysis only conducted part way through? This raises concerns for me, e.g., whether the analyses were run with just this initial wave of data collection, before more data were collected when results were not significant. However, given the lack of statistically significant findings in the final sample, it seems irrelevant to ask for any earlier results to be reported (although they could be added to OSF, of course, for transparency, if they exist).
Thanks for this question. We did not conduct an a-priori power analysis because this is a new task and it would have been impossible to estimate the effect size without any prior data available. Of course, we could have used “standard” effect sizes (e.g., small, medium, large) to run a priori analysis before collecting any data, but we believe this information would not have been very useful, particularly because it is very difficult to estimate variability due to items a priori. In contrast, our approach involved collecting a first wave of data, then using simulations to estimate how many more participants would be needed to achieve a desired level of power given the observed effect size (and observed levels of item and participant variability). First and second wave of data collection are clearly labelled as such on OSF, so interested readers can reproduce our power analyses and re-run our analysis only on the first wave of data if they wish.
- I’m not really sure what the following sentence is saying: “When words were repeated, we made sure to counterbalance which role (theme/recipient) they played in the sentences and whether they played the same or a different role on their second occurrence.” (lines 257-9) Did these repeated nouns always appear in different roles? It’s not clear to me.
Thanks for this request for clarification. On lines 261-264, we have added the following to remove any ambiguity.
This means that even numbers of repeated words played the same role across repetitions or switched roles (and if the latter, theme ïƒ recipient or recipient ïƒ theme switches were equally likely).
- In frequentist statistics, results are either significant or not. Please therefore rephrase “marginal interaction” (line 394) and “a marginal effect of predictability” (lines 411-12), e.g., to describe these as trends. Similarly, lines 437-8 imply that an effect was found, whereas the stats don’t support this. Have the authors considered running Bayesian analyses instead? This would likely be of benefit in terms of the conclusions that can be drawn.
Thanks for catching our sloppy phrasing here! We have removed all uses of the term “marginal”. We have also carefully reviewed the rest of our results and discussion sections to make sure we do not imply that an effect was found at any point. Thanks for suggesting Bayesian analyses as an alternative but we do not think they would allow us to draw any stronger conclusions than we can draw on the basis of the frequentist analysis. As we acknowledge in the manuscript, we think the study is underpowered and the between-participant variability too high– because of this we suspect the Bayesian analysis would simply show the empirical evidence is not conclusive one way or the other, which we already acknowledge by stressing our results need to be followed up with more (and better) data.
Reviewer 3 Report
Comments and Suggestions for Authors
This article tests error-driven models of language learning for L1 acquisition of English syntactic structures, specifically direct object datives. The article tested the hypothesis that 4-year-old children would learn how to interpret difficult syntactic structures in English when they process input that facilitates prediction errors. „Acquisition“ or „learning“ is defined as improvement on the comprehension of syntactic structures in the post-test compared to the pre-training testing phase. The authors report on a study that was underpowered according to their own assessment and thus provide preliminary conclusions which are based on numerical differences observed. The authors call for additional research to provide stronger empirical support for or against the hypotheses tested.
Overall, the topic of the article is very timely and a great fit for the special issue, the article is well-structured and written well (but could use clarification in some points, see below). The article focuses on the comprehension-side rather than the production side, which brings a nice new angle to the discussion. The authors point out the limitations of their study throughout. The article and experiment appear worthwhile to be published despite its shortcomings.
There are a lot of methodological and statistical details that are both highly useful and relevant, though can be slightly confusing especially given the limitations the authors address throughout. I would therefore suggest that the authors clarify the following points to make the article easier to comprehend for readers:
1. Be clearer about which part(s) of the statistics and the ensuing discussions provide support for the claim that generating incorrect predictions led to more learning.
2. Data analysis: aren’t both c) predictability and d) trial difficulty related to the Animate vs. Inanimate opposition? At least this is how it sounds in the description on page 3. Please clarify whether these can be considered separate factors in the later data analyses. (I realize that Animacy was not predicted to make a difference for the results, but I am still confused as to whether c) and d) can be completely separate factors. It’s possible that they are separable - if so, it would be good to clarify this issue at the beginning of the data analysis section.)
3. It is not clear to me what exactly „experimental errors“ are (p. 7-8), i.e. experimenter responses on the tablet. How could there have been up to 7% errors? (line 349) This seems high. But again, I don’t think I understand what is meant by this or how to imagine what happened in these situations. Please clarify.
4. The discussion of Figure 2 does not seem to be easy to follow.
Methodologically, can the authors describe the prosody (prosodic contours) used in the recorded sentences and whether or how this was controlled for the different syntactic structures (DO vs PO) during the recordings?
Finally, can the authors discuss whether these effects might be merely task-effects rather than true learning or acquisition? Without a delayed post-test, can we be sure of any lasting effects? Even with a delayed post-test, can we know whether this type of experiment reveals more than practice effects within this particular task as opposed to (true) learning or acquisition of how to comprehend the syntactic structures in (spoken) language?
